# Estimation of Soybean Internal Quality Based on Improved Support Vector Regression Based on the Sparrow Search Algorithm Applying Hyperspectral Reflectance and Chemometric Calibrations

**Kezhu Tan \*, Qi Liu, Xi Chen, Haonan Xia and Shouao Yao**

Electrical Engineering and Information College, Northeast Agriculture University, Harbin 150030, China; lq03131999@163.com (Q.L.); chenxi08050121@163.com (X.C.); llzlbzg@163.com (H.X.); 15656341863@163.com (S.Y.)
* Correspondence: z80053@neau.edu.cn

**Abstract:** The nutritional components of soybean, such as fat and protein, directly decide soybean quality. The fast and accurate detection of these components is significant to soybean industries and soybean crop breeding. This study developed an improved SSA-SVM (support vector regression based on the sparrow search algorithm) for the rapid and accurate detection of the fat and protein in soybean seeds using hyperspectral reflectance data. In this work, 85 soybean samples were selected. After their fat and protein contents were analyzed using chemical methods, a total of 85 groups of hyperspectral image data were collected using the hyperspectral imaging system. An effective data preprocessing method was applied to reduce the noise for enhancing the prediction models. Some popular models, including partial least-square regression (PLSR), random forest regression (RFR), and support vector regression based on the genetic algorithm (GA-SVR), were also established in this study. The experimental results showed that the improved SSA-SVM model could predict the nutrient contents of the soybean samples with accuracies of 0.9403 and 0.9215 and RMSEs of 0.2234 and 0.325 for the fat and protein, respectively. The convergence speed was improved significantly. Therefore, hyperspectral data combined with the SSA-SVM algorithm presented in this study were effective for evaluating the soybean quality.

**Keywords:** SSA-SVM model; fast nondestructive detection; hyperspectral image technology

## 1. Introduction

Soybean is one of the most important food crops in the whole world [1]. It is rich in many kinds of nutrition [2], such as unsaturated fatty acids and amino acids, which are necessary for the human body. In particular, soybean stands out for its higher levels of fat and protein, making it a primary source of plant-based oil and natural protein. Consequently, the fat and protein contents serve as critical indicators in the assessment of soybean quality. However, conventional chemical techniques used to measure protein and fat have some drawbacks, including their time consumption, labor intensiveness, destructive nature, environmental pollution, and methodological complexity [3]. Hence, the development of rapid, nondestructive testing technology has become imperative for accurately predicting soybean protein and fat contents.

In recent years, there has been notable progress in the development of spectroscopic techniques. Near-infrared spectroscopy, hailed for its simplicity, rapidity, environmental friendliness, and nondestructive nature [4], has emerged as a popular method for the analysis and detection of the chemical composition, quality, and characteristics of a variety of samples [5,6]. However, the measurement methodology can impact the accuracy of the obtained data, presenting certain limitations in the use of near-infrared spectroscopy. Hyperspectral imaging captures continuous spectral information across the visible and

near-infrared spectrum, typically consisting of hundreds or even thousands of narrow bands [7]. This approach provides a wealth of spectral data [8], with each band serving as an independent spectral channel capable of capturing subtle differences in and features of the target object. Compared to near-infrared spectroscopy, hyperspectral imaging can provide a broader wavelength coverage, a higher spectral resolution, and more abundant information retrieval, enabling more accurate analysis and prediction capabilities. Hyperspectral technology has been applied in various fields [9], including agriculture [10], the food industry [11], meat inspection [12], biomedicine [13], and soil science [14–16]. It plays a crucial role in substance identification, classification, and quantitative analysis.

When using hyperspectral technology, extracting the most effective bands from a vast array of variables is a crucial step in enhancing the predictive capabilities of models [17]. Many researchers have dedicated their efforts to reducing the dimensionality of the data by eliminating redundant information, irrelevant variables, and noise [18]. To achieve optimal band extraction, researchers often compare the accuracies of multiple dimensionality reduction algorithms across different applications. There are several popular algorithms for feature band selection, including the Successive Projection Algorithm [19] (SPA), Uninformed Variable Elimination [20] (UVE), Step-Wise Regression [21] (SWR), and Competitive Adaptive Reweighted Sampling [22] (CARS). These algorithms serve the purpose of identifying the most informative and relevant bands, enabling researchers to focus on the most significant features for their specific applications. Najmeh Haghbin [23] employed CARS, UVE, and the SPA to select wavelengths with maximum information content for studying the effects of gray-mold infection on the hardness, soluble solid content, and titratable acidity attributes of kiwifruit. In order to further reduce the complexity of the model and improve its accuracy, some scholars have proposed a combination of multiple dimensionality reduction methods to select important variables. In the prediction of the antioxidant activity in osmanthus flowers, Zhou F [24] established the optimal antioxidant activity model, UVE-SPA-MLR. Compared to using separate UVE and SPA filtering methods, the combination UVE-SPA shows higher accuracy.

The support vector machine (SVM) regression algorithm has a strong generalization ability and robustness, and it is especially suitable for predicting regression problems with limited sample sizes. Eleni Kalopesa [25] used SVR (support vector machine regression) to estimate the sugar content in wine grapes. When using SVR, it is necessary to consider the optimization problem of the C and g parameters to control the risk of overfitting and the lack of generalization ability [26]. A grid search [27] is a straightforward and commonly used strategy for parameter optimization. However, searching within a given combination may lead to insufficient search accuracy. In addition, although a random search covers the parameter range with the desired accuracy, it significantly increases the computation cost and time. Meta-heuristic algorithms have gained popularity due to their ability to quickly search for optimal solutions across the entire parameter space, striking a balance between the two aforementioned strategies. The genetic algorithm (GA) applied by Liu Q [28], particle swarm optimization (PSO) studied by Ni Zifan [29], and other algorithms further optimize parameters such as C and g through repeated iterations, resulting in more accurate and faster prediction results. In 2020, Xue et al. [30] introduced a new and efficient swarm intelligence optimization algorithm called the sparrow search algorithm (SSA), inspired by the foraging and anti-predation behaviors of sparrows. The SSA provides a fresh perspective for solving optimization problems. Paul V [31] utilized the SSA to optimize a mixed short-term and long-term memory model for water quality prediction. However, during the optimization process, challenges such as becoming stuck in local extreme points early on and low precision in the later stages may arise. To address these issues, Tang Yanqiang [32] proposed the strategies of adaptive adjustment and mutation perturbation, significantly improving the optimization of the mathematical benchmark functions.

This study uses the CARS-SPA algorithm to extract relevant features, and a ZSYSSA-SVM model is established for predicting soybean protein and fat content. The specific objectives are as follows:

(1) Propose the CARS-SPA combination algorithm to filter out redundant characteristic wavelengths of soybeans.
(2) Propose a kind of SSA algorithm to optimize the prediction model of soybean protein and fat.
(3) Propose three improvement strategies for their impact on SSA and evaluate their impact on soybean protein and fat prediction models.
(4) Propose the best soybean protein and fat prediction model and analyze its application.

## 2. Materials and Methods

### 2.1. Materials and Experiments

Heilongjiang Province is the largest soybean production region in China. In 2022, its soybean planting area reached 492.67 thousand square hectares, accounting for 48.1% of the national total. Its soybean production reached 9.54 billion kilograms, accounting for 47% of the national total, ranking first in the country. In order to make the experiment more representative, samples were selected from the five major soybean production areas in Heilongjiang Province: Heihe, Qiqihar, Jiamusi, Suihua, and Harbin. Figure 1 shows the soybean collection area. The soybeans were harvested uniformly on 30 September 2021. After four days of air drying, they were stored in a low-temperature environment. For this experiment, 85 soybean varieties were selected, including Jiami 12, Jiami 13, Heihe 33, and Heihe 43. Each group of samples consisted of 100 uniformly sized and undamaged soybeans.

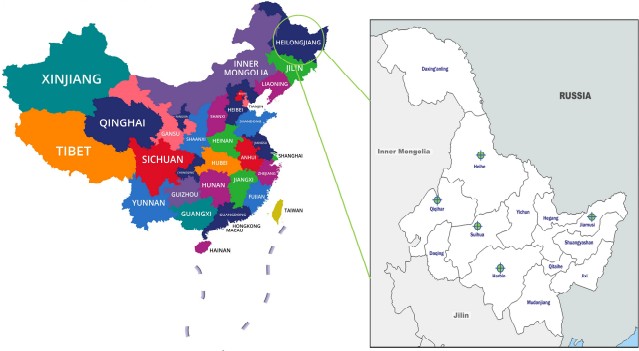

**Figure 1.** Distribution map of soybean sample production areas.

Following the GB5009.5-2021 standard, we employed the Kjeldahl method to determine the protein content in soybeans. Initially, we accurately weighed a specific weight of soybean samples. These samples were combined with concentrated sulfuric acid and an appropriate catalyst in a Kjeldahl digestion tube, followed by heating to convert the organic nitrogen in the sample into ammonium sulfate. Once the digestion process was complete, the digest was cooled and then slowly neutralized with a sodium hydroxide solution, converting the ammonium sulfate into ammonia gas. Through distillation, the ammonia gas was separated from the mixture and absorbed using a boric acid solution of known concentration. Subsequently, the acid solution that absorbed the ammonia gas was titrated with a standard sodium hydroxide solution until the endpoint was reached, allowing the amount of acid consumed to be determined. Based on the amount of acid consumed, the total nitrogen content in the sample was calculated. Then, using a protein conversion factor (typically 6.25), the protein content in the soybean sample was determined. Following the Soxhlet extraction method provided in the GB5009.6-2016 standard for measuring soybean fat, we ground the soybean sample into a fine powder, accurately weighed a specific amount of this powder, and placed it into an extraction thimble. We inserted the thimble with the sample into a Soxhlet extractor and added an adequate amount of a solvent, such

as ether or petroleum ether, into the extractor's flask. We heated the solvent in the flask until it evaporated, allowing it to rise, condense, and then drip into the thimble containing the sample. The solvent dissolved the fat in the sample and was then evaporated by heating, condensed back, and recycled through the extraction process. After a set extraction period, we ceased heating, removed the thimble, and allowed all the solvent to evaporate completely. We placed the thimble in an oven and dried it to a constant weight to eliminate any residual solvent. We weighed the dried thimble and residue again. The difference in weight before and after was the amount of fat in the sample. To ensure the accuracy of the physicochemical property results, each sample was measured three times, and the average of these three measurements was taken as the final data for the physicochemical properties. The minimum percentage content of protein among all samples was 38.8%, the maximum one was 46.7%, the average one was 42.38%, and the standard deviation was 2.09%. The minimum percentage content of the fat among all samples was 18.7%, the maximum one was 22.5%, the average one was 20.95%, and the standard deviation was 1.07%.

## 2.2. Hyperspectral Data Acquisition and Correction

The experiment was conducted in the Hyperspectral Image Processing Laboratory at the College of Electrical and Information Technology, Northeast Agricultural University. The hardware system included the Hyperspec®VNIR hyperspectral imager, which consists of a camera with a CCD (charge-coupled device) image sensor and a spectrometer with a VNIR (visible/near-infrared) spectral range, as well as a motor-controlled transport platform and a halogen lamp. The hyperspectral images were collected by Hyperspec III software, with a spectral range of 400.92–999.53 nm. After interference factors were removed from the raw data, the spectral range was 463–957 nm, with a bandwidth of a 1 nm interval. A total of 495 spectral bands were collected. During the sample collection, firstly, the instrument was preheated for 20–30 min to ensure a stable light source. In order to meet the requirement of maintaining good stability for the soybeans throughout the testing process, the soybeans were evenly placed in a self-made 10 × 10 mobile platform with a completely black background. Then, the hyperspectral images were acquired at a speed of 3.6 mm/s. The range for sample manipulation was set from 30 mm to 180 mm. If an obvious distortion occurred during this process, a remeasurement was performed. The experimental procedure is illustrated in Figure 2.

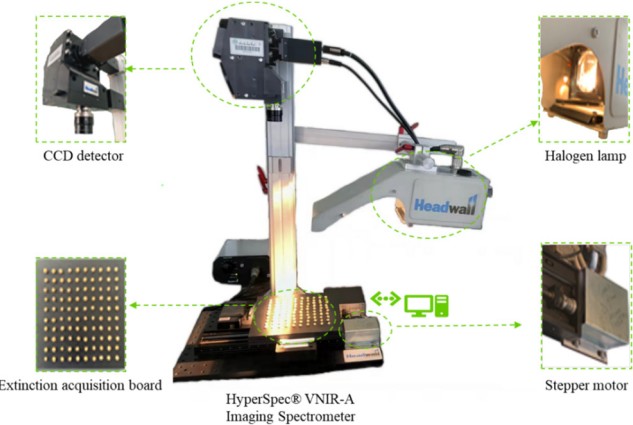

**Figure 2.** Hyperspectral experiment process.

The data extraction from the sample image was performed using ENVI 5.3. Within each group of samples, 100 regions of interest (ROIs) were sequentially delineated, as illustrated in Figure 3. We calculated the average of these ROIs and used the results as raw data. The raw spectral curve, depicted in Figure 4, was plotted based on these data. Simultaneously, the corresponding data from the black and white reference panels were also extracted.

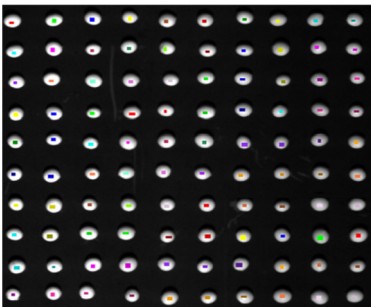

**Figure 3.** Soybean spectral ROI extraction.

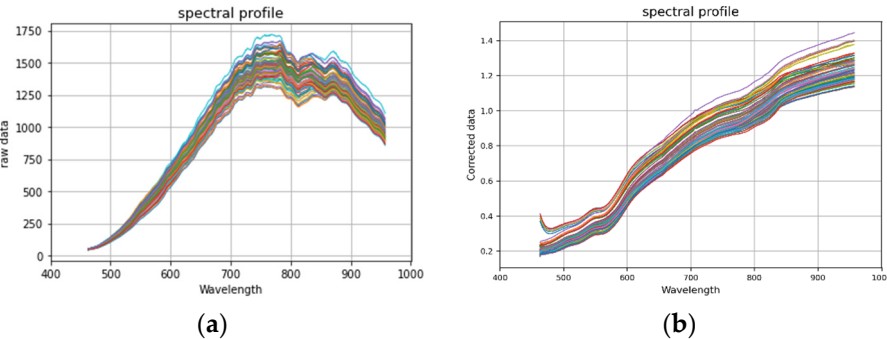

(**a**)                                    (**b**)

**Figure 4.** Spectral curves of soybean samples: (**a**) original spectral curves of soybean samples; (**b**) spectral curve of soybean sample after correction.

Factors such as dark current and uneven illumination from the light source have a certain impact on collected spectral images. Therefore, corrected images were estimated by using the following equation:

$$R_j = \frac{R_y - R_h}{R_b - R_h} \tag{1}$$

where $R_j$ is corrected spectral data, $R_y$ is raw spectral data, $R_b$ is the white reference data, and $R_h$ is the dark reference data. The data in the subsequent experiments were all from the corrected spectral data. The following figure shows the spectral data curves before and after calibration, (a) showing the curves of the band and instrument response values and (b) showing the curves of the band and calibrated spectral data.

### 2.3. Data Processing

In order to increase the difference between spectra, improve the identification rate, and eliminate the interference of the baseline and other factors, here, we introduced data preprocessing, including the center norm, first derivative SG, multiple scatter correction (MSC), and standard normal variate (SNV). At the same time, the problems of high dimensionality and high data volume need to be solved. If the data were used directly, there would be overfitting, and the effect of the verification would become low. Therefore, it was necessary to reduce the dimension of the data, remove the bands with low correlation, and express the original spectral information with fewer dimensions to improve the performance of the model. In this experiment, the methods of SPA, CARS, and CARSSPA were applied for feature selection.

### 2.4. Model Establishment

The quality of the model has a decisive impact on the prediction of the fat and protein. Here, we established three prediction models, including PLSR, RFR, and SVR, and used GA and SSA to optimize the parameters of 'C' and 'g'.

*2.5. Sparrow Search Algorithm and Its Improved Strategy*

2.5.1. SSA

In 2020, Xue and Shen of Donghua University proposed the sparrow search algorithm inspired by the predatory and anti-predatory behaviors of common sparrows in life. In the process of foraging, sparrows are divided into discoverers, joiners, and scouts. Generally, it is believed that there is a relationship between competition and cooperation among sparrows. The discoverers are responsible for searching for the direction and area with sufficient food in the process of foraging, and they have high energy reserves. The joiners are often found in an environment with low energy and poor foraging positions within the population. They will follow the discoverer to seize food resources with high energy reserves and improve their fitness. When the scouts find the existence of predators, they will issue a warning message, and then the sparrows at the edge of danger will move to a safe area. The identities of the three roles may be exchanged at any time to obtain food more safely and efficiently.

First, set the number of sparrows and the boundary range of sparrows' activities.

$$X = \begin{bmatrix} x_1^1 & x_1^2 & \cdots & x_1^d \\ x_2^1 & x_2^2 & \cdots & x_2^d \\ \cdots & \cdots & \cdots & \cdots \\ x_n^1 & x_n^2 & \cdots & x_n^d \end{bmatrix} \tag{2}$$

where d is the dimension and n is the number of sparrows.

Second, use the evaluation index RMSE of SVR to set the fitness function, sort the fitness value, and select the optimal value.

$$\text{fitness} = \text{argmin}(\text{RMESTrain} + \text{RMSETest}) \tag{3}$$

Then, update the position of the discoverers in the sparrows, the joiners in the sparrows, and forecasters in the sparrows.

$$X_{i,j}^{t+1} = \begin{cases} X_{i,j} \cdot \exp\left(-\frac{i}{\alpha \cdot \text{iter}_{max}}\right), & \text{if } R_2 < ST \\ X_{i,j} + Q \cdot L, & \text{if } R_2 \geq ST \end{cases} \tag{4}$$

where t is the current number of iterations, $X_{i,j}$ is the location information of the i sparrow in dimension j, $\alpha$ is a random number in (0, 1], $\text{iter}_{max}$ is the maximum number of iterations, $R_2$ is the value of the danger coefficient, ST is the safety critical value, and Q is a random number in the normal distribution.

$$X_{i,j}^{t+1} = \begin{cases} Q \cdot \exp\left(\frac{X_{worst} - X_{i,j}^t}{i^2}\right), & \text{if } i > n/2 \\ X_{best}^{t+1} + \left|X_{i,j} - X_{best}^{t+1}\right| \cdot A^+ \cdot L, & \text{otherwise} \end{cases} \tag{5}$$

where $X_{worst}$ is the global worst position, $X_{best}$ indicates the global best location, and $A^+$ expresses $A^T(AA^T)^{-1}$.

$$X_{i,j}^{t+1} = \begin{cases} X_{best}^t + \beta \cdot \left|X_{i,j}^t - X_{best}^t\right|, & \text{if } f_i > f_g \\ X_{i,j}^t + K \cdot \left(\frac{\left|X_{i,j}^t - X_{worst}^t\right|}{(f_i - f_w) + \varepsilon}\right), & \text{if } f_i = f_g \end{cases} \tag{6}$$

where K is a random number in [−1, 1]; $f_i$ is the fitness value of a single sparrow; $f_w$ is the worst fitness value of a current sparrow individual; $f_g$ is the best global fitness value at present; $\varepsilon$ is a very small number, which guarantees that the denominator is not zero; and $\beta$ is a random number in the (0, 1) normal distribution that is used to control the step size.

Next, update the sparrow position iteratively and calculate the fitness value of each position.

Lastly, at the end of the iteration process, find the minimum fitness value in the whole world, which corresponds to the optimal solution of the objective function, namely the optimal C and g. The specific SSA-SVM parameter optimization process is shown in Figure 5.

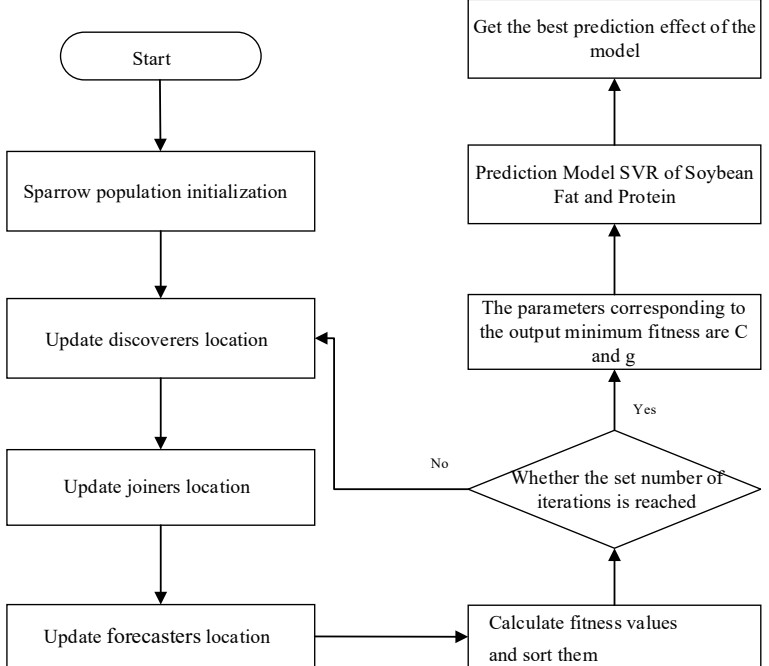

**Figure 5.** SSA-SVM parameter optimization process.

2.5.2. Improved Strategies

Strategy 1: chaos initialization population of cat mapping

In order to improve the ergodicity, randomness, and global search ability of the initial population, we change the random initial population into a chaotic initial population. In this experiment, the cat mapping chaotic strategy is used to generate the initial population of the sparrow search algorithm. The cat mapping expression is

$$\begin{bmatrix} y_{i+1} \\ w_{i+1} \end{bmatrix} = \begin{bmatrix} 1 & a_1 \\ b_1 & a_1 b_1 + 1 \end{bmatrix} \begin{bmatrix} y_i \\ w_i \end{bmatrix} \bmod 1 \tag{7}$$

where $a_1$ and $b_1$ are arbitrary real numbers; Mod1 is the decimal part of $a_1$.

The initialization process of generating a chaotic sequence within the specified activity limit is as follows:

Randomly generate a solution for the sparrow population:

$$\{Y_i = [y_{i1}, y_{i2}, \dots, y_{id}, \dots, y_{iD}]; y_{id} \in [lb_{id}, ub_{is}]\} \tag{8}$$

The reverse solution is

$$Y' = [Y'_1, Y'_2, \dots, Y'_d, \dots, Y'_D] \tag{9}$$

The initial population of cat mapping is

$$y_{id} = q(lb_{id} - ub_{id}) - y_{id} \tag{10}$$

where q is evenly distributed in [0, 1], and $lb_{id}$ and $ub_{id}$ represent the active boundary of sparrows.

Strategy 2: Tent chaos and Cauchy variation disturbance

In order to jump out of the local optimum and enhance the global search ability, we add some disturbances in the sparrow search process. Tent chaos and Cauchy mutation were used in this experiment.

In order to improve the feature of the Tent chaotic map falling into small and unstable periods, a random variable is introduced here. The chaotic perturbation formula is obtained as follows:

$$z_{i+1} = \begin{cases} 2z_i + \text{rand}(0,1) \times \frac{1}{N} & 0 \leq z \leq \frac{1}{2} \\ 2(1-z_i) + \text{rand}(0,1) \times \frac{1}{N} & \frac{1}{2} < z \leq 1 \end{cases} \tag{11}$$

In order to reduce the peak value at zero, ensuring it decreases slowly towards zero and achieves a more uniform range, we modify the original Cauchy distribution. The variation formula is

$$\text{mutation}(x) = x(1 + \tan(\pi(u - 0.5))) \tag{12}$$

where x is the position of the previous sparrow, mutation(x) is the position of the mutated sparrow, and μ is a random number in (0, 1).

Strategy 3: discoverers' and joiners' adaptive adjustment strategy

In order to improve the accuracy and convergence speed of the algorithm, we added the sparrows' internal adaptive adjustment process. In this experiment, the discoverers and the joiners adjust to each other. At the beginning of the iteration, more discoverers conduct a global search within the active boundary to speed up the convergence; in the middle and late stages of the iteration, as the scope shrinks, the discoverers adaptively convert to joiners and adjust the global search to a local search to improve the accuracy of the optimization.

The discoverer and joiner adaptive adjustment is as follows:

$$r = b\left(\tan\left(-\frac{\pi t}{4 \cdot \text{iter}_{max}} + \frac{\pi}{4}\right) - k \cdot \text{rand}(0,1)\right) \tag{13}$$

where b is a proportional coefficient that controls the number relationship between the discoverers and the joiners; K is the disturbance coefficient.

After adaptive adjustment, the number of the discoverers becomes

$$j = r \cdot N \tag{14}$$

After adaptive adjustment, the number of the joiners becomes

$$j = (1 - r) \cdot N \tag{15}$$

## 3. Results

### 3.1. Spectral Pretreatment Analysis

To remove noise interference, enhance the spectral fitting effect, and ensure data validity, the preprocessing of the corrected spectral data was performed in this experiment. The PLS regression model was utilized to evaluate different pretreatment methods, with evaluation indicators including $R_p^2$, RMESP, $R_c^2$, and RMSEC. Table 1 presents the results, indicating that the fat models showed improvement after undergoing pretreatments such as the center norm, first derivative, MSC, and SNV. Among these methods, the center norm model exhibited the highest Rc value of 0.8633 and the lowest RMSE of 0.3698. Notably, this model's scores were 2.55%, 2.62%, and 7.92%, respectively, higher than those of the other three models. For the protein, the center norm method also yielded the highest Rc value of 0.8706, accompanied by the lowest RMSE of 0.8334. As a result, the center norm was selected as the preferred preprocessing method for subsequent data applications. Figure 6 illustrates the spectral curves of the four pretreatment methods (Additional reference and verification are in Supplementary Materials).

**Table 1.** Comparison of pretreatment models.

| Object | Pretreatment Method | Train Set | | Test Set | |
|---|---|---|---|---|---|
| | | Rc$^2$ | RMSEC | Rp$^2$ | RMSEP |
| fat | raw data | 0.5731 | 0.6713 | 0.3133 | 0.9826 |
| | Center Norm | 0.9509 | 0.2423 | 0.8633 | 0.3698 |
| | First Derivative | 0.8698 | 0.3946 | 0.8378 | 0.4027 |
| | MSC | 0.8499 | 0.4238 | 0.8371 | 0.4036 |
| | SNV | 0.9983 | 0.0447 | 0.7841 | 0.4364 |
| protein | raw data | 0.5337 | 1.3615 | 0.329 | 1.8984 |
| | Center Norm | 0.9676 | 0.3588 | 0.8706 | 0.8334 |
| | First Derivative | 0.9318 | 0.5204 | 0.8354 | 0.9402 |
| | MSC | 0.8671 | 0.7266 | 0.842 | 0.921 |
| | SNV | 0.9998 | 0.0265 | 0.8546 | 0.8836 |

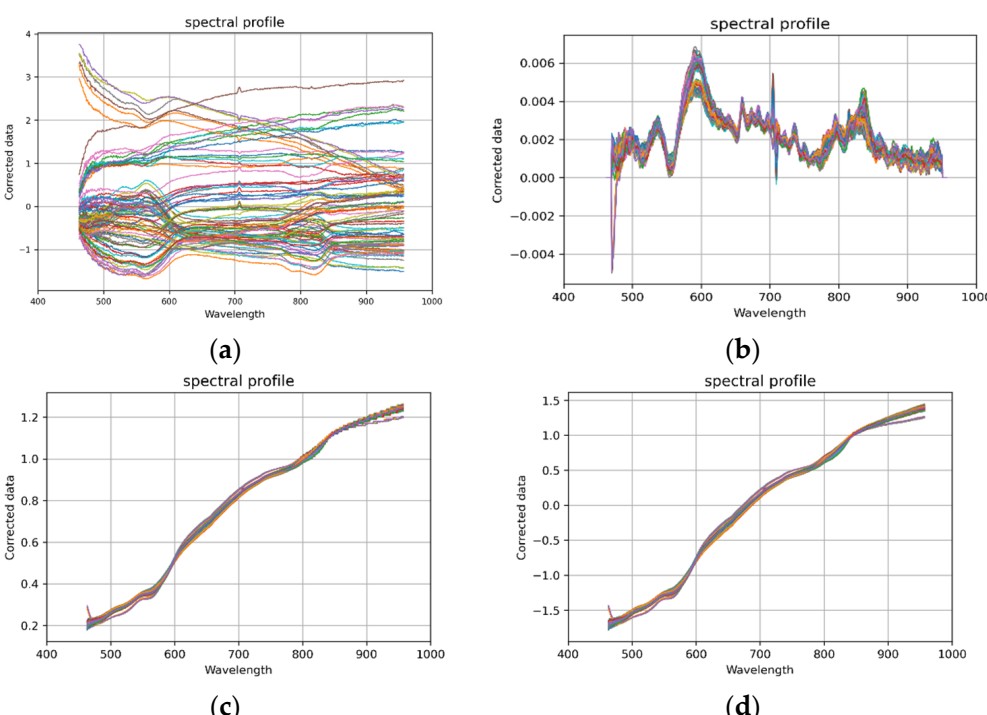

**(a)**

**(b)**

**(c)**

**(d)**

**Figure 6.** Pretreatment spectral curve. (**a**) Center norm. (**b**) First derivative. (**c**) MSC. (**d**) SNV.

*3.2. Analysis of Feature Selection Scheme*

To enhance the accuracy, adaptability, and robustness of the model, it is crucial to eliminate irrelevant information from the data while retaining the characteristic wavelengths that have a significant impact on the target variables. In this study, three algorithms were employed for band selection: CARS, SPA, and CARSSPA. These algorithms aim to identify the most informative and relevant spectral bands, thereby improving the overall performance of the model.

3.2.1. Feature Selection of CARS

CARS is a feature selection method based on the principle of "survival of the fittest". In the initial stage, an exponential decay function is used to quickly select and eliminate irrelevant variables. In the middle and later stages, a refined selection process is conducted using the exponential decline function. The performance of the feature selection is evaluated by establishing a PLS regression prediction model and calculating the Root Mean Square Error of Cross-Validation (RMSECV). The optimal combination of bands is determined

when the RMSECV reaches its minimum value. In this study, the number of Monte Carlo samples was set to 50.

Figure 7 illustrates the CARS feature selection process for both fat and protein. As the number of iterations increases, the corresponding RMSECV first decreases and then increases. The red vertical line in the figure indicates the minimum number of iterations. For the fat, when the number of iterations was 14, the RMSECV reached its minimum value of 0.3179, with a corresponding optimal combination of 103 bands. The coefficient of determination ($R^2$) for the fat reached 0.879. On the other hand, for the protein, the RMSECV was minimized at the 23rd iteration, with the smallest RMSE and a combination of 37 features. The corresponding $R_p^2$ for the protein was 0.8518. The CARS feature selection process for the fat and protein is presented in Figure 7, and the distribution in the original spectrogram is depicted in Figure 8.

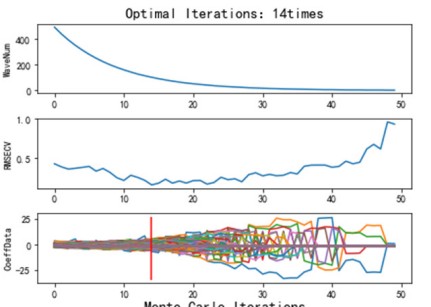 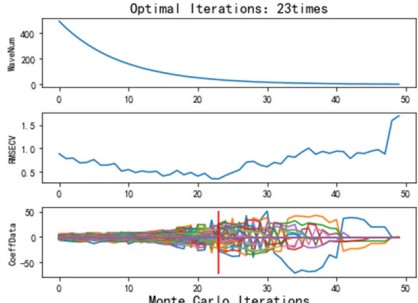

**Figure 7.** CARS feature selection of fat and protein.

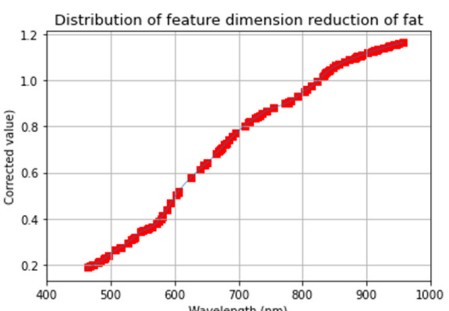 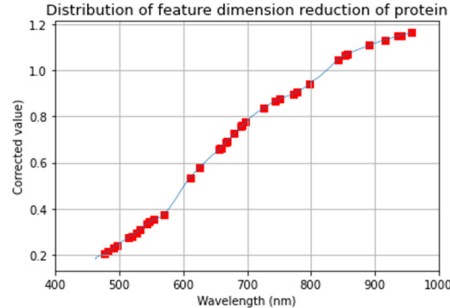

**Figure 8.** Mapping of CARS results in spectrograms.

3.2.2. Feature Selection of SPA

SPA is an algorithm that employs forward search to minimize collinearity. It calculates the projection values of different spectral bands to screen and retain the best combination of wavelengths, effectively reducing redundant information, simplifying the model, and improving operational efficiency.

Figure 9 illustrates the screening process for the fat and protein variables using SPA. The RMSE initially decreases, then increases, and eventually stabilizes. For the fat variable, the minimum RMSE of 0.3191 was obtained when the number of variables was 14. The distribution of the selected characteristic variables is shown in Figure 10. Similarly, for the protein feature selection, the RMSE reached its minimum value of 0.7325 when the number of variables was 13. Remarkably, the band selection process resulted in compressing the variables to only 2.6% of the full band.

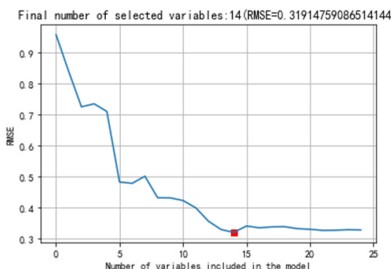
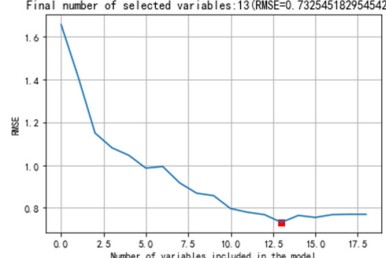

**Figure 9.** SPA feature selection of fat and protein.

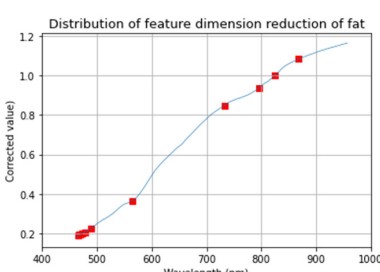
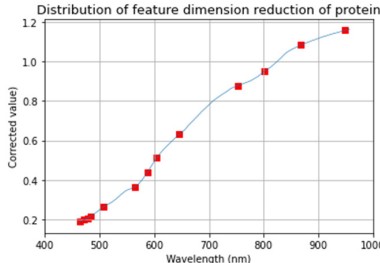

**Figure 10.** Mapping of SPA results in spectrograms.

### 3.2.3. Feature Selection of CARSSPA

After the features were screened with CARS, the number of bands still presented more information, while SPA could effectively compress redundant variables. Therefore, the joint feature extraction of SPA based on CARS could not only ensure the retention of effective features, but also minimize variables and simplify the model.

Figure 11 shows the process of screening the fat and protein variables using the CARSSPA algorithm. The RMSE in the PLSR model is used as the evaluation index. For the fat variable, the RMSE reached its minimum value of 0.2685 when the number of variables was 24. Similarly, for the protein variable, the RMSE achieved its minimum value of 0.4123 with 19 characteristic bands. The distribution of the selected feature variables is shown in Figure 12.

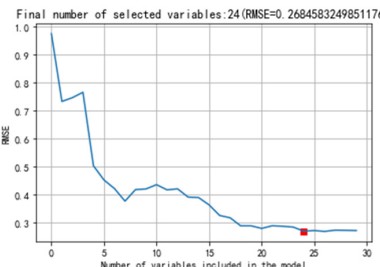
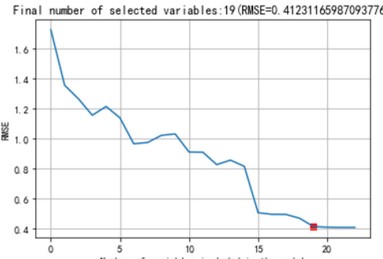

**Figure 11.** CARSSPA feature selection of fat and protein.

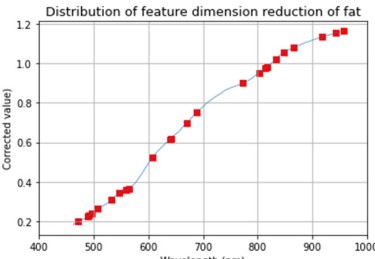
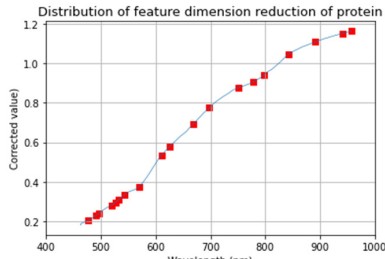

**Figure 12.** CARSSPA feature selection variable distribution chart.

The results obtained from CARS, SPA, and CARSSPA feature selection methods are compared in the SVM model, as shown in Table 2.

**Table 2.** Comparison of feature selection models.

| Object | Pretreatment Method | Number of Features | Train Set | | Test Set | |
|---|---|---|---|---|---|---|
| | | | $Rc^2$ | RMSEC | $Rp^2$ | RMSEP |
| fat | CARS | 103 | 0.8617 | 0.3717 | 0.879 | 0.3179 |
| | SPA | 14 | 0.783 | 0.4657 | 0.8079 | 0.4007 |
| | CARSSPA | 24 | 0.8812 | 0.3445 | 0.901 | 0.2876 |
| protein | CARS | 37 | 0.9465 | 0.2312 | 0.8798 | 0.4073 |
| | SPA | 13 | 0.8499 | 0.3873 | 0.8467 | 0.455 |
| | CARSSPA | 19 | 0.9018 | 0.3133 | 0.8966 | 0.3737 |

For the fat variable, CARSSPA extracted 24 bands, achieving an $Rp^2$ of 0.901. Compared to CARS, CARSSPA demonstrated a compression rate of 76.7%, an improvement in $Rp^2$ by 2.2%, and a reduction in RMSE by 0.0303. Additionally, CARSSPA compensated for some relevant features that were eliminated by SPA, resulting in a 9.31% increase in the $Rp^2$ rate. Regarding the protein variable, CARSSPA retained 19 bands with an $Rp^2$ of 0.8966. Compared to CARS and SPA, CARSSPA exhibited an increase in $Rp^2$ by 0.0448 and 0.0499, respectively. Moreover, the RMSE for CARSSPA decreased by 0.0736 and 0.0813, respectively, compared to CARS and SPA. These improvements highlight the significant enhancement in the model performance achieved by CARSSPA.

Based on these results, CARSSPA was selected as the preferred feature selection method for subsequent analysis. It effectively compressed the number of variables while improving the performance of the SVM model for both fat and protein content predictions in soybeans.

*3.3. Analysis of Prediction Model*

PLSR, RFR, and SVR are widely used in traditional regression prediction models in various fields such as food, medicine, and agriculture. In this experiment, a PLS regression model was employed to predict the fat and protein contents. The parameter 'n_components' was optimized using grid search. The maximum $Rp^2$ values for the fat and protein contents were achieved when 'n_components' was set to 16 and 13, respectively, resulting in $Rp^2$ values reaching 0.9309 and 0.8946. For the prediction using the random forest regression (RFR) model, a search was conducted to find the optimal combination of parameters from three given arrays. For fat content prediction, the parameters 'max_depth', 'max_features', and 'n_estimators' were set to 15, 0.7, and 10, respectively. For protein content prediction, they were set to 5, 0.6, and 20, respectively. The performance of the RFR model reached its best with prediction effects of the fat and protein contents reaching 0.9063 and 0.8645, respectively, as shown in Table 3.

**Table 3.** Comparison of regression prediction models.

| Object | Method | Train Set | | Test Set | |
|---|---|---|---|---|---|
| | | $Rc^2$ | RMSEC | $Rp^2$ | RMSEP |
| fat | PLSR | 0.9646 | 0.2056 | 0.9309 | 0.2628 |
| | RFR | 0.9673 | 0.1976 | 0.9063 | 0.3061 |
| | SVR | 0.8812 | 0.3445 | 0.901 | 0.2876 |
| | GA-SVR | 0.8988 | 0.318 | 0.9359 | 0.2314 |
| | SSA-SVR | 0.9349 | 0.255 | 0.9396 | 0.2246 |
| | ZSYSSA-SVR | 0.9342 | 0.2564 | 0.9402 | 0.2235 |
| protein | PLSR | 0.9609 | 0.4002 | 0.8946 | 0.7231 |
| | RFR | 0.9539 | 0.4278 | 0.8645 | 0.8531 |
| | SVR | 0.9018 | 0.3133 | 0.8966 | 0.3737 |
| | GA-SVR | 0.9301 | 0.2642 | 0.9152 | 0.3383 |
| | SSA-SVR | 0.9294 | 0.2656 | 0.9213 | 0.326 |
| | ZSYSSA-SVR | 0.9295 | 0.2655 | 0.9215 | 0.3256 |

In this experiment, three optimization algorithms, namely GA, SSA, and an improved algorithm called ZSYSSA, were used to optimize the parameters 'C' and 'g' of the SVR model. The optimization process employed the RMSE as the fitness function, with an initial population of 50 and a maximum of 100 iterations. The parameter bounds were set as [0.001, 100]. We repeated the experiment to take the mean value for analysis.

The adjustment of parameters C and g by three optimization algorithms has improved the performance of the model. For the fat variable, the improved SSA optimization of SVR achieved a 3.92% increase in $Rp^2$ compared to ordinary SVR, along with a decrease in RMSE by 0.0641. The optimal values for C and g were found to be 1.75577165 and 0.3684816, respectively. Compared with GA and SSA, the $Rp^2$ increased by 0.43 and 0.06, respectively. Regarding the protein variable, the ZSYSSA optimization algorithm yielded the best results, with the optimal value of 'C' being 30.68672218 and that of 'g' being 0.16808313. The optimized $Rp^2$ values were ranked as ZSYSSA > SSA > GA, and the RMSE values were ranked as ZSYSSA < SSA < GA. Although the improvement in $Rp^2$ with ZSYSSA was not significantly different from SSA, Figure 13 demonstrates that the optimal solution was achieved with fewer iterations, indicating a faster convergence speed compared to SSA.

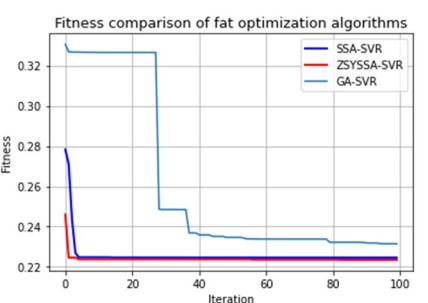 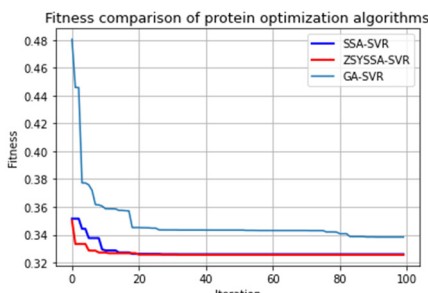

**Figure 13.** Optimization algorithm fitness function curve.

These findings suggest that the ZSYSSA optimization algorithm outperformed GA and SSA in terms of convergence speed and model performance. It provides an efficient approach for optimizing the SVR model for predicting the fat and protein contents in soybeans.

The deviation degree between the actual value and the predicted value could well reflect the fitting effect of the model. The following figure shows the regression of the fat and protein under different models; the closer the regression line was with respect to y = x, the better the performance is. Figures 14 and 15 show the deviation degree between the actual value and the predicted value of the fat and protein.

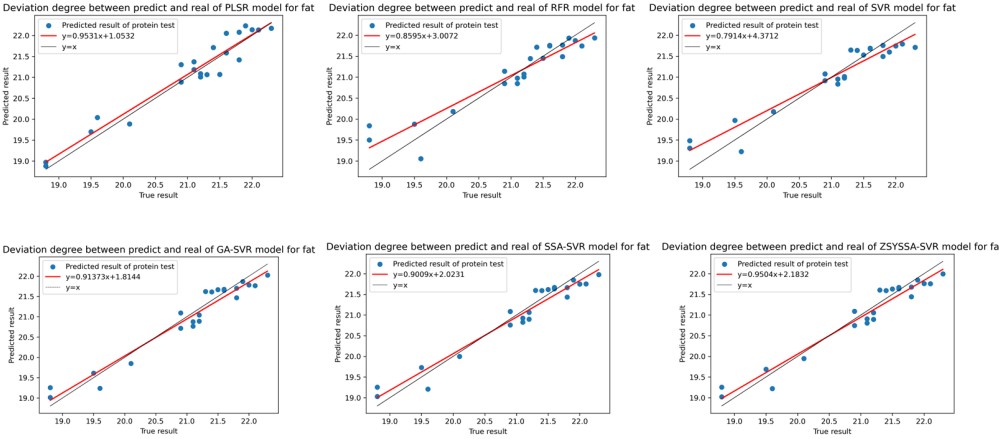

**Figure 14.** Deviation degree between true value and predicted value of fat.

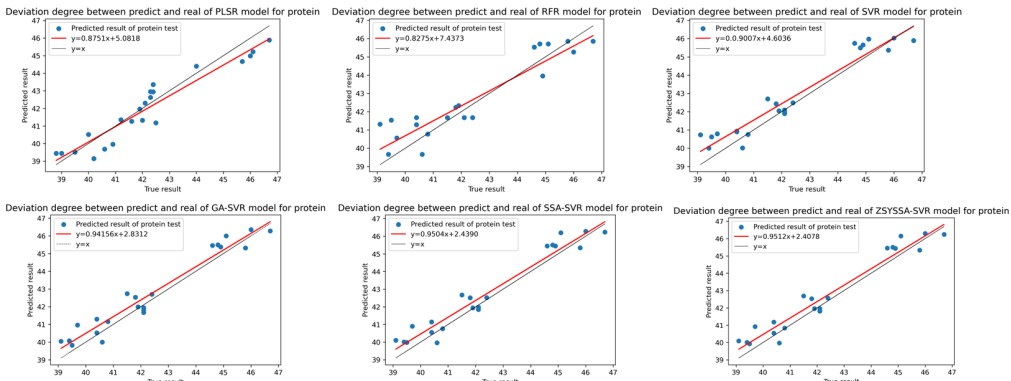

**Figure 15.** Deviation degree between true value and predicted value of protein.

### 3.4. Best Model Selection

Through the exploration of this experiment, it has been found that employing center norm for data pretreatment, CARSSPA for feature selection, SVM for model building, and ZSYSSA for further model improvement can yield good results in predicting the fat and protein contents of soybeans.

Table 4 presents the determination coefficients for the fat and protein, which reached 0.9402 and 0.9215, respectively. These coefficients indicate the accuracy and reliability of the model in predicting the fat and protein contents. Figure 16 provides a visualization of the results achieved by the optimized model. The figure illustrates the predicted values compared to the actual values for the fat and protein, highlighting the closeness of the regression lines to the ideal line of y = x. This visual representation further supports the effectiveness of the developed model in predicting the soybean fat and protein contents.

**Table 4.** Optimal regression model for predicting soy protein and fat content.

| Object | Method | Train Set | | Test Set | |
|---|---|---|---|---|---|
| | | $R_c^2$ | RMSEC | $R_p^2$ | RMSEP |
| fat | CARSSPA-center norm-ZSYSSA-SVR | 0.9342 | 0.2564 | 0.9402 | 0.2235 |
| protein | CARSSPA-center norm-ZSYSSA-SVR | 0.9295 | 0.2655 | 0.9215 | 0.3256 |

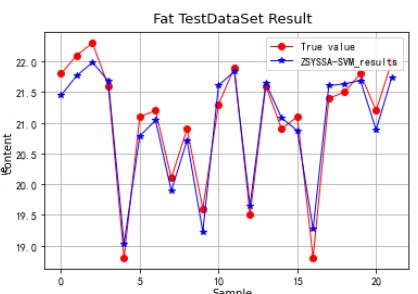 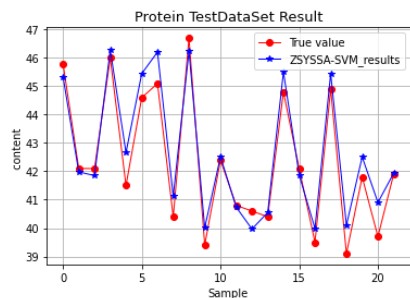

**Figure 16.** Best model fitting effect.

The combination of center norm pretreatment, CARSSPA feature selection, SVM model optimization, and ZSYSSA model improvement demonstrates promising results and can serve as a reliable approach for the nondestructive prediction of soybean fat and protein.

### 4. Discussion

In this study, we demonstrated the significant advantage of the CARSSPA algorithm, which integrates CARS and SPA, in dimensionality reduction. Through the competitive adaptive reweighted sampling by CARS and stepwise information feature selection by SPA, this algorithm effectively eliminates redundant features while retaining essential

information, thereby substantially improving the accuracy and efficiency of model predictions. The applicability of PLSR, RFR, and SVR models for predicting the soybean fat and protein contents was evaluated, confirming their effectiveness. Additionally, the study highlights the importance of parameter optimization in enhancing the model performance, showcasing the superiority of heuristic search algorithms over the traditional grid search in optimizing predictive performance. Particularly noteworthy is the heuristic optimization algorithm inspired by the foraging behavior of sparrows used in this research, which, through the chaos initialization, interference enhancement during the search process, and an adaptive adjustment mechanism, significantly speeds up the model convergence during training, enhances the optimization efficiency, and further improves the accuracy of the fat and protein content predictions.

Moreover, we believe that this methodology is not only applicable to the detection of fat and protein in soybeans but also has broader application prospects. Given the capability of hyperspectral imaging technology to capture subtle differences, we anticipate that this combined algorithm could also be applied to the analysis of other foodstuffs and their components (sugars, starch, fiber, vitamins, minerals, etc.). The flexibility and high level of information retrieval offered by this technology present new possibilities for food safety and quality control, especially in scenarios requiring rapid, nondestructive testing. Future research could further explore the practicality of this method in detecting other foods and components, as well as further algorithm optimization to enhance its accuracy and reliability in practical applications.

## 5. Conclusions

This article, utilizing the hyperspectral technology in conjunction with the improved SSA-SVM model, predicts the protein and fat contents in soybeans, yielding the following significant conclusions: firstly, the combination algorithm based on CARS-SPA is more effective for dimensionality reduction; secondly, SSA performs better in optimizing SVR's C and g compared to the traditional grid search and genetic search algorithms; furthermore, the SSA based on the above three improvement strategies achieved the best results; lastly, choosing the CARS-SPA-ZSYSSA-SVR model can achieve the best predictive values for soybean protein and fat content.

Further, compared to the traditional chemical methods, this improved SSA-SVM method exhibits significant advantages such as ease of operation, rapid response, and nondestructiveness, which are crucial for enhancing experimental efficiency and preserving sample integrity. Additionally, the method offers high-precision and high-efficiency predictions, which is vital for precise control in food processing and quality assessment. However, there are certain limitations, including the dependence on hyperspectral equipment and complexity in data processing, which may increase experimental costs and build technical barriers. With technological advancements and optimizations, this hyperspectral and machine learning-based approach will play an increasingly important role in food safety and agricultural product quality control.

**Supplementary Materials:** The following supporting information can be downloaded at: https://www.mdpi.com/article/10.3390/agriculture14030410/s1.

**Author Contributions:** Resources, Supervision, Funding Acquisition, K.T.; Conceptualization, Methodology, Software, Validation, Writing—Original Draft, Q.L.; Validation, Investigation, X.C.; Methodology, Validation, H.X.; Methodology, S.Y. All authors have read and agreed to the published version of the manuscript.

**Funding:** This work was supported by the Heilongjiang Provincial Natural Science Foundation, grant number LH2020C003.

**Institutional Review Board Statement:** Not applicable.

**Data Availability Statement:** The raw data supporting the conclusions of this article will be made available by the authors on request.

**Acknowledgments:** The authors fully appreciate the editors and all anonymous reviewers for their constructive comments on this manuscript.

**Conflicts of Interest:** The authors declare no conflict of interest.

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
