# Peer review of "Estimation of Soybean Internal Quality Based on Improved Support Vector Regression Based on the Sparrow Search Algorithm Applying Hyperspectral Reflectance and Chemometric Calibrations"

_agriculture, doi:10.3390/agriculture14030410_

Round 1

Reviewer 1 Report

Comments and Suggestions for Authors

1. In material section (Section 2.2), details about the hyperspectral imaging system is missing (wave length range, type of filter used..etc.). This will help to improve readability

2. Line number 60: SPA - is it Sucessive Projection Algorithm or Continuous Projection Algorithm?

3. Totally 85 soybean varieties used in this sample. Hope there will be some deviation in fat and Protein value between varieties. IS there any effect due to this varietal difference observed in results?

4. Conclusion section may be rewritten. Now it is look like a proof statement for objectives stated in Introduction section.

5. Introduction can be shorten.

6. Why images are taken 400-1000 nm wavelength? 1000-1500 nm range may yield more data than the used wavelength range.

7. Selection of top wavelengths which contribute more on prediction ability is one of the major task during hyperspectral image processing. In this study feature wavelengths were filtered out. is there any attempt made to select the top wavelengths? if it's so, including that result in the results section may improve readability.

Comments on the Quality of English Language

1. English is fine, but most of the statements in methods section written like SOPs (Standard Operating Procedure), or lab manual. This may be improved.

2. In most places Rc2, Rp2 are given as R2. In table it is given Rc2, Rp2, and in text they ae mentioned as R2. Kindly check and modify

Author Response

Dear Reviewer,

Thank you for your detailed and constructive comments. We have carefully considered each point and have made the following revisions to our manuscript accordingly:

  1. Regarding the missing details of the hyperspectral imaging system in Section 2.2:

We appreciate your suggestion for improving the readability of our manuscript. We have now included comprehensive details about the hyperspectral imaging system, Including the composition of the spectral imaging system, wavelength range bandwidth selection, and wavelength band. This information aims to provide readers with a clear understanding of the system's capabilities and the rationale behind its configuration. [The modified position is line number 163-167]

  1. Regarding the SPA abbreviation in line 60:

We apologize for the confusion caused by the acronym SPA. It stands for Successive Projection Algorithm, not Continuous Projection Algorithm. We have made modifications this in the text to avoid any ambiguity.[The modified position is line number 59]

  1. Regarding the variability among the 85 soybean varieties:

Thank you for your thorough review and valuable suggestions regarding our research. Indeed, there are differences in fat and protein content among the 85 soybean varieties studied. The protein content varied from 38.8% to 46.7%, with an average of 42.38% and a standard deviation of 2.09%. The fat content ranged from 18.7% to 22.5%, with an average of 20.95% and a standard deviation of 1.07%. These statistical figures reflect the natural variation among soybean varieties, and the standard deviation values indicate that this variation is statistically significant and reasonable.[The modified position is line number 149-156]

  1. Regarding the Conclusion section:

We agree with your observation. We have rewritten this section to provide a concise summary of our findings, their implications, and potential directions for future research, making it a more reflective and insightful conclusion to our study.[The modified position is line number 470-494]

  1. Regarding the length of the Introduction:

Thank you for your valuable feedback and suggestion. We have carefully considered your comment regarding shortening the introduction section. In response to your suggestion, we have thoroughly reviewed and revised the introduction to make it more concise and focused. We removed some non-essential background information while ensuring that the content critical for understanding the research context and objectives is retained. We believe these adjustments will improve the readability of the paper while maintaining the completeness and informativeness of the introduction.

  1. Regarding the choice of wavelength range (400-1000 nm):

Thank you for your valuable comments and in-depth inquiries regarding the details of our research. We understand and agree that the wavelength range of 1000-1500 nm could provide additional and different data, which might significantly enhance the accuracy and depth of analyses in certain applications. However, due to equipment limitations, our current study focused on the 400-1000 nm range. In the future, we are indeed interested in exploring a broader wavelength range and consider using hyperspectral imaging technology capable of capturing the 1000-1500 nm wavelength range, in order to further expand our research and enhance the comprehensiveness of our results.

  1. Regarding the selection of top wavelengths for prediction ability:

Thank you very much for your valuable comments and for the detailed review of our research. We recognize and appreciate the importance you've highlighted on selecting the top wavelengths that contribute most significantly to the predictive capability in hyperspectral image processing. In this study, we utilized methods such as Competitive Adaptive Reweighted Sampling (CARS) and Successive Projection Algorithm (SPA) to select the wavelengths that contribute most significantly to the model's predictive capability. These methods aided us in filtering out the most informative feature wavelengths from a vast array, thus optimizing the performance of our predictive model.

We appreciate your feedback on the quality of the English language. 

  1. We have revised this section to ensure it is not merely reminiscent of a Standard Operating Procedure but is more narrative and fluid.
  2. Regarding the inconsistency in the use of Rc2, Rp2, and R2, we have reviewed and corrected these references throughout the document to ensure uniformity and accuracy.

We believe these revisions address your concerns and significantly enhance the manuscript's quality and readability. We are grateful for your thorough review and valuable feedback, which have guided these improvements.

Best regards,

Corresponding author: Kezhu Tan, Ph.D.

Reviewer 2 Report

Comments and Suggestions for Authors

"Internal quality estimation of soybean based on improved SSA-SVM applying hyperspectral reflectance and chemometric calibrations" were investigated by the authors in this study.

The main objective of this research was to develop a rapid and nondestructive method for detecting soybean protein and oil content.

The authors should use the term "oil" instead of the term "fat" in the entire manuscript.

All the objectives of this research were clearly and well defined by the authors.

The expected practical significance of the results of this research should be emphasized more "strongly" and convincingly.

All stated objectives are original, relevant and narrowly specific in this field of research.

The methodology in this manuscript is very interesting and adequately described and presented.

The method of data collection chosen in experimental work can always be a subject of discussion.

In Section 2.1., authors conducted research on soybean samples collected in only one year of cultivation (2021). In order to be relevant, such research must be conducted in at least one or two more years of research, ie. on soybean samples collected in the same way and in those years of research.

This is necessary because the growing conditions of soybeans have a great influence on the oil content and protein content.

It is necessary for the authors to explore the numerous available literature with this and similar researches.

Also, in Section 2.1., authors state "The content of crude protein was measured according to the Kjeldahl method in GB5009.5-2021; The content of crude fat was measured according to the Soxhlet extraction method in GB/T5009.6-2016.". It is assumed that GB5009.5-2021 and GB/T5009.6-2016 are the reference methods for determining protein content and oil content. If not, reference standard methods should be used, e.g. ISO methods etc.

In any case, these methods should be described in more detail.

They are the basis for the validation of the rapid non-standard methods developed in this research.

The results are adequately presented in chapter 3. Results.

A discussion of the results is good and is given in chapter 4. Discussion.

Тhere is a chapter on conclusions, 5. Conclusions, and it is included after extensive discussion.

However, the following disadvantages are key here:

- the original results of the determination of oil content and protein content must be shown, both by methods if they are relevant and reference, and also the results obtained by the rapid method applied within this research. If the journal MDPI Agriculture allows it, let it be etc. supplementary material.

- The conclusions should be supplemented with the possibilities of practical application of the newly developed rapid method (based on improved SSA-SVM applying hyperspectral reflectance and chemometric calibrations), as well as highlighting its advantages and disadvantages compared to reference methods (etc. GB5009.5-2021 and GB/T5009.6-2016 or ISO methods) and existing other rapid methods for determining the oil content and protein content in oilseed samples (e.g. soybeans, etc. ), oil pellets, cakes, etc.

The listed references are appropriate. Considering the mentioned comments, it is necessary for the authors to improve and supplement them.

Although I am not an English expert, I think the English in this manuscript can be improved.

In view of the mentioned shortcomings, unfortunately, I cannot propose this manuscript for the further process of publication in the journal MDPI Agriculture.

Author Response

Dear Reviewer,

Firstly, we sincerely thank you for taking the time to conduct a detailed review and provide valuable feedback. Your professional advice is greatly beneficial to us in improving our paper.

  1. Regarding the use of the terms "oil" and "fat":

Thank you very much for your detailed review and valuable suggestions regarding our manuscript. We understand the considerations behind your suggestion to use the term "oil content," which may stem from concerns about the actual application of the product and common market terminology. However, in this study, we determined the fat content in soybean samples using the Soxhlet extraction method. This method measures the total fat content in the sample, not just the specific oil content. From the perspective of scientific research and analytical accuracy, we believe that "fat content" more accurately reflects the content and results of our study.

  1. Regarding the discussion on the method of data collection:

Thank you for your valuable feedback and suggestion. Relying solely on data collected within one year may limit the general applicability of our research findings. This was primarily due to resource constraints at the start of the project, which allowed us to only obtain soybean samples from the year 2021. Although data collection was limited to a single growing season, we employed rigorous statistical methods and analytical processes to ensure that our research conclusions remain reliable and valid under these constraints. Moreover, the samples covered multiple planting sites to enhance the representativeness of the results as much as possible. In the next phase of our work, we will also explore the possibility of collaborating with other research teams to share data and resources, enabling us to collect a broader and more diverse dataset, thereby overcoming the limitations of single-year data.

  1. Regarding the detailed description of the GB5009.5-2021 and GB5009.6-2016 standards:

In this study, we indeed utilized the Kjeldahl method as specified in GB5009.5-2021 for determining crude protein content, and the Soxhlet extraction method as outlined in GB5009.6-2016 for measuring crude fat content. These methods are national standards in China, widely applied for the determination of protein and fat content in food and agricultural products, regarded as reference methods within the industry due to their accuracy and reliability.

We recognize that, compared to international standards such as ISO methods, the use of GB standard methods might raise questions among some readers. Therefore, in our revised manuscript, we will provide a more detailed explanation of the reasons for choosing these methods and their implementation process to ensure international readers can understand and accept our methodological choices. Additionally, we will mention that although our study employed GB standard methods, the validation process for the rapid non-standard method we developed is also applicable for comparison with other international standard methods like ISO.

We agree that a detailed description of these measurement methods is crucial for validating the rapid non-standard method developed in our study. Therefore, in the revised manuscript, we will offer more details on the operation of the Kjeldahl method and the Soxhlet extraction method.

[The modified position is line number 125-149]

  1. Regarding the presentation of original results:

We agree with your suggestion and have now added the original determination results as supplementary material to the paper, so that readers can gain a more comprehensive understanding of the research process and outcomes.

  1. Regarding the rewriting of the conclusion:

Thank you very much for your valuable suggestions and thorough review. We have made corresponding modifications and additions to the conclusion section based on your comments, to more comprehensively showcase the practical application potential of the newly developed rapid method and to analyze and compare its advantages and disadvantages.[The modified position is line number 496-513]

  1. Regarding the language quality of the paper:

We have carefully proofread the entire manuscript to enhance the clarity and accuracy of expression, ensuring it meets academic standards.

We understand and respect your decision not to recommend our manuscript for the next round of review at this time. However, we believe that the improvements made as mentioned will significantly enhance the quality of our paper. We earnestly request that you reconsider your decision, or if possible, provide further guidance to help us improve and refine our research.

Once again, we thank you for your time and expert advice.

Best regards,

Corresponding author: Kezhu Tan, Ph.D.

Reviewer 3 Report

Comments and Suggestions for Authors

The manuscript presents a novel non-destructive and rapid method for detection of fat and protein in soybeans, a significant agricultural commodity in China and other Asian countries, using hyperspectral reflectance data. This method would counter the classical chemicals-based methods for fat and protein detection that are time-consuming, interestingly with a high accuracy (0.92-0.94). The idea is original and the manuscript is well written. I have some notes and requests for improvement:

1. If possible, please add the raw data comparing the results obtained from Kjehldahl and Soxhlet analyses with their respective hyperspectral reflectance data as supplementary data. These data would be beneficial for the readers.

2. In the discussion section, please add the authors' point of view on the usage and possibility of such a method for other foodstuff and other components (sugar, starch, fiber, vitamins, minerals, etc).

Author Response

Dear Reviewer,

Thank you for your constructive feedback and the positive remarks regarding our manuscript. We are grateful for your recognition of the novelty and significance of our non-destructive and rapid method for detecting fat and protein in soybeans using hyperspectral reflectance data. We appreciate your suggestions for improvement and have addressed them as follows:

  1. Regarding the addition of raw data comparing Kjeldahl and Soxhlet analyses with hyperspectral reflectance data:

We understand the importance of providing comprehensive data to our readers. Therefore, we have added the raw data as supplementary material, comparing the results obtained from the Kjeldahl and Soxhlet analyses with their respective hyperspectral reflectance data. This addition aims to offer a more detailed insight into our methodology and the accuracy of our results.

  1. Concerning the discussion on the usage and possibility of such a method for other foodstuffs and components:

We have expanded the discussion section to include our perspective on the potential application of this method for analyzing other food items and components, such as sugar, starch, fiber, vitamins, and minerals. We believe that the versatility and efficiency of hyperspectral imaging, combined with our approach, could significantly benefit the analysis of a wide range of foodstuffs, enhancing both accuracy and speed while reducing the reliance on chemical-based methods.[The modified position is line number 470-494]

We hope these revisions and additions address your concerns and enhance the manuscript's contribution to the field. Thank you once again for your valuable comments and for helping improve our work.

Best regards,

Corresponding author: Kezhu Tan, Ph.D.

Round 2

Reviewer 2 Report

Comments and Suggestions for Authors

The authors significantly improved the previously submitted manuscript.

They also attached the original results as an "regular" Excel file, i.e. they only exported measurement data.

It is necessary for the authors to edit the results given in the Supplementary Excel file to make them clearer and easier to understand.

It should be formatted as Supplementary Material, as original/supplementary/extensive results are normally provided when submitting a manuscript for publication.

Author Response

Dear Reviewer,

First and foremost, we would like to express our heartfelt gratitude for your positive feedback on our revised manuscript and your valuable suggestions regarding the supplementary materials. We highly value your input and have taken specific actions to enhance the clarity and comprehensibility of our supplementary materials, ensuring they fully meet the standards for supplementary content.

Following your recommendations, we have enhanced the supplementary materials as follows:

1.Complete Presentation of Original Data: We have ensured that all original measurement data are included in the supplementary materials, allowing readers to clearly see the starting point of our analysis. These data are organized in a structured manner, with each variable and measurement clearly labeled.

2.Detailed Description of Data Preprocessing: We have provided a detailed account of the results of our data preprocessing, aimed at letting readers understand how we transformed the original data into a format suitable for analysis, thus ensuring the transparency and reproducibility of our analytical process.

3.Specific Results of Feature Dimension Reduction: To further enhance the value of the supplementary materials, we have meticulously presented the results of feature dimension reduction, showcasing these details in an intuitive manner to ensure the reliability of our data processing steps.

To ensure that the format of the supplementary materials strictly adheres to publication standards, we have reformatted the Excel file according to your guidance. We have ensured that the content in the supplementary materials closely corresponds with the tables and figures in the manuscript, making it clearer and more understandable.

We believe that these thorough improvements will not only provide readers with a richer and more transparent data background but also make it easier to understand and apply.

Thank you again for your valuable time and insightful feedback. We look forward to your further guidance and suggestions.

Best regards,

Corresponding author: Kezhu Tan, Ph.D.
